

# Toxic dinoflagellate *Prorocentrum cordatum* affects the filtration rate and enzymatic activities of Chinese razor clam (*Sinonovacula constricta*)

Yanbin Tang[1,2,3,4], Zhibing Jiang[2,3,4], Yibo Liao[2,3,4], Lu Shou[2,3,4], Jiangning Zeng[2,3,4], Rongliang Zhang[2,3,4] and Chenghua Li[1]

[1] School of Marine Sciences, Ningbo University, Ningbo, China
[2] Key Laboratory of Marine Ecosystem Dynamics, Second Institute of Oceanography, Ministry of Natural Resources, Hangzhou, China
[3] Key Laboratory of Nearshore Engineering Environment and Ecological Security of Zhejiang Province, Hangzhou, China
[4] Observation and Research Station of Yangtze River Delta Marine Ecosystems, Ministry of Natural Resources, Zhoushan, China

Corresponding author
Chenghua Li,
lichenghua@nbu.edu.cn

## ABSTRACT

Harmful algal blooms represent a significant environmental challenge in various marine ecosystems worldwide. While marine filter-feeder bivalves can consume toxic phytoplankton, their capacity to mitigate the presence of harmful microalgae is not yet fully understood. In this study, we examined the filtration rates and enzymatic activities of *Sinonovacula constricta*, a commercially valuable bivalve, when exposed to varying levels of toxic dinoflagellates (*Prorocentrum cordatum*) and non-toxic diatoms (*Skeletonema costatum*) over a 12-h period. Chlorophyll *a* concentration was used to reflect the presence of these microalgae. In the initial 2 h, the filtration rate under toxic conditions was lower than under non-toxic conditions. However, after the first 2 h, the filtration rate under toxic conditions did not decline as rapidly as it did under non-toxic conditions, suggesting that *S. constricta* could adapt to the presence of toxic microalgae over time. Regarding enzymatic activities, digestive enzymes were not significantly affected by low concentrations of toxic microalgae, but lipase activity was inhibited at higher concentrations. Antioxidant enzyme activity showed no significant changes across all non-toxic microalgal concentrations. Superoxide dismutase (SOD) activity increased at higher toxic microalgal concentrations, but both low SOD and catalase activities indicated that the bivalve's antioxidant defenses for detoxification may be limited. These results suggest that *S. constricta* can tolerate toxic microalgae through adaptive feeding behaviors and changes in digestive and antioxidant enzymatic activities. This study revealed *S. constricta* has a high filtration rate and is sensitive to high concentrations of toxic microalgae. Therefore, its bioremediation function requires further study.

## INTRODUCTION

Harmful algal blooms (HABs) in marine environments, also called red tides, are often caused by environmental changes that demonstrate the expanding global human footprint and effects of climate change (*Stauffer et al., 2019*; *Zohdi & Abbaspour, 2019*). HABs involve multiple species and classes of microalgae that produce toxins or other bioactive substances that adversely affect beneficial aquatic organisms by influencing disease susceptibility or death by predation or parasitism, resulting in community structure alteration and lead to broader, potentially undesirable changes in habitat (*Zohdi & Abbaspour, 2019*; *Young et al., 2020*). Recurrent HABs plague coastal waters worldwide and their frequency is predicted to increase (*Hallegraeff, 2010*), owing to coastal eutrophication and warming (*Paerl et al., 2016*; *Stauffer et al., 2019*). Many environmental factors, such as chlorophyll *a* have been applied as indicators of photoautotrophic biomass as related to primary productivity for monitoring the HABs (*Boyer et al., 2009*). The dinoflagellate *Prorocentrum cordatum* (Ostenfeld) Dodge 1976 (former name: *Prorocentrum minimum* (Pavillard) Schiller 1933) is one of the major bloom-forming species in warm, temperate coastal waters around the world (*Velikova & Larsen, 1999*; *Khanaychenko, Telesh & Skarlato, 2019*). *P. cordatum* is known to produce diarrhetic shellfish poison (DSP) and cause fish and shellfish mortality, thus posing a serious risk to aquaculture species and human health (*Azanza et al., 2005*; *Sahraoui et al., 2013*).

Filter-feeder bivalves consume phytoplankton, including toxic species. Some studies have reported that dinoflagellate toxicity can affect bivalve feeding behavior and associated physiological processes (*de Romero-Geraldo, García-Lagunas & Hernández-Saavedra, 2016*; *Zohdi & Abbaspour, 2019*). The filtration rate of bivalves is mainly influenced by microalgal species and density (*Bayne et al., 1993*). Bivalves can reduce their filtration rate by withdrawing their siphons and/or closing their shells to resist high-density toxic microalgae (*Bardouil et al., 1996*; *Istomina et al., 2021*). Toxin accumulation in bivalves occurs in the digestive gland and affects digestive enzymatic activities (*Vidal et al., 2014*). Most bivalves can transform and transport toxins through a series of acylation and hydrolysis processes catalyzed by antioxidant enzymes such as superoxide dismutase (SOD) and catalase (CAT) (*Vidal et al., 2014*; *Istomina et al., 2021*; *Tan et al., 2022*). Bivalves can exhibit varying degrees of tolerance or resistance to harmful algal blooms. Bivalve species may respond to toxic cells by rejecting them (*Bauder et al., 2001*; *Rosa et al., 2017*) or by ingesting the cells and subsequently eliminating the associated toxins (*Blanco, Estévez-Calvar & Martín, 2025*). This tolerance often involves physiological and cellular adaptations (*Lassudrie et al., 2020*).

The Chinese razor clam (*Sinonovacula constricta* Lamarck 1818) is a common, benthic filter-feeding bivalve that is widely distributed along the coast of the western Pacific Ocean (*Orita et al., 2021*; *Yao et al., 2021*). *S. constricta* is an economically important aquaculture clam species in China. The clams usually half-bury themselves in the soft bottom of mudflats, reaching out their siphons to filter water and consume microalgae and suspended organic particles in the water. *S. constricta* can also improve water quality by filtering suspended solids and reducing nutrient fluxes in the water body (*Yang et al., 2017*;

*Zhao et al., 2019*). Therefore, *S. constricta* has been used as a bioremediation species in the aquaculture industry to improve polluted water (*Zhao et al., 2019*; *Zhang et al., 2022*). Although *S. constricta* has great potential to reduce levels of harmful algae, the response of the clam to toxic microalgae has not been reported.

The primary aim of this study was to determine the short-term effects of toxic dinoflagellate *P. cordatum* on the feeding behavior and associated physiological processes of *S. constricta*. To achieve this goal, the filtration rate and activities of four digestive enzymes and two antioxidant enzymes of *S. constricta* were explored with different concentrations of *P. cordatum* compared to those under nontoxic conditions with diatom *Skeletonema costatum* (Greville) Cleve 1873. The results of this study provide evidence for evaluating the suitability of *S. constricta* as a biological mitigation agent for harmful algal blooms, by elucidating its capacity to maintain filtration efficiency and physiological stability under exposure to toxic *P. cordatum*, and thereby its potential effectiveness in reducing algal biomass in natural waters.

## MATERIALS AND METHODS

### Microalgae culture and clam collection

*Prorocentrum cordatum* and *Skeletonema costatum* were aseptically maintained in f/2 medium (*Guillard & Ryther, 1962*) in 10 L glass conical flasks at 22 °C and 100 μmol photon $m^{-2} s^{-1}$, with a 12:12 h light/dark cycle. The chlorophyll *a* content of each conical flask was measured daily using a Hawk TriLux fluorometer (Chelsea Technologies Ltd., West Molesey, UK). When the chlorophyll *a* concentration reached over 200 μg $L^{-1}$, the microalgae were dispersed into four new conical flasks to reduce microalgal mortality caused by high density.

*Sinonovacula constricta* were collected from a bivalve aquaculture farm in Xiangshan, Zhejiang, China. One-year-old adult calms with similar lengths (674.2 ± 4.6 mm) and weights (25.2 ± 3.6 g) were selected and divided into two groups: exposed and control. Each group was placed in a polypropylene carbonate tank filled with seawater for five days and was fed *S. costatum* daily to allow the clams to adapt to the laboratory environment. Dead clams and feces were removed from the tanks daily, which were then filled with clean seawater.

### Exposure experiment and filtration rate

After the 5-day adaptation period, 24 clams that could extend and retract their siphons normally were selected from each group and starved one day before the exposure experiment. The cultivated microalgae were diluted with filtered seawater to arrive at the four chlorophyll *a* concentration treatments. Each treatment consisted of six replicate tanks each containing one clam and two tanks containing no clams. All tanks were filled with 2 L filtered seawater mixed with *P. cordatum* (exposed group) or *S. costatum* (control group). The exposed and control groups each consisted of four treatments according to the chlorophyll *a* concentration of the microalgae: 60.2 ± 0.8 μg $L^{-1}$ (treatment 1), 126.8 ± 1.4 μg $L^{-1}$ (treatment 2), 171.9 ± 1.9 μg $L^{-1}$ (treatment 3), and 229.2 ± 2.5 μg $L^{-1}$

(treatment 4). The chlorophyll *a* concentration in treatment 1 was slightly above the typical threshold for HABs (40 μg L$^{-1}$) (*Busari, Sahoo & Jana, 2024*), while treatment 2 represented a strong HABs event commonly observed along the Chinese coast (*Chen et al., 2023*). To further investigate the tolerance of *S. constricta* to toxic microalgae, two additional treatments (treatment 3 and 4) were designed with chlorophyll *a* concentrations exceeding those found in most HABs. The water temperature was 23.1 ± 0.6 °C and salinity was 28.0 ± 0.5 during the exposure experiment. *S. constricta* has a diurnal cycle rhythm of feeding rate that is highly associated with digestive enzyme activities (*Liu et al., 2021*). To minimize the influence of the diurnal cycle upon the filtration rate and enzymatic activities, the exposure experiment lasted for 12 h (from 6 am to 6 pm). No clam mortality was observed during the exposure experiment. From 0 to 6 h, the concentration of chlorophyll *a* was measured every hour. From 6 to 12 h, the chlorophyll *a* concentration was measured every 2 h. All chlorophyll *a* concentrations throughout the exposure experiment were measured by a Hawk TriLux fluorometer (Chelsea Technologies Ltd., West Molesey, UK).

The filtration rate (FR) of the clams was calculated for each group based on chlorophyll *a* concentration and expressed in unit μg h$^{-1}$. The FR can be expressed as follows:

$$FR = \frac{V[(A_0 - A_1) - (B_0 - B_1)]}{T} \tag{1}$$

where, $V$ is the volume of seawater, $A_0$ is the initial chlorophyll *a* concentration in the tank with clams, $A_1$ is the chlorophyll *a* concentration after time $T$ (h), $B_0$ is the initial chlorophyll *a* concentration in the tank without clams, and $B_1$ is the chlorophyll *a* concentration after time $T$ (h).

## Determination of enzymatic activities

After the exposure experiment, clams were dissected immediately, and the digestive glands were separated, individually homogenized in saline solution, and centrifuged at 2,500 rpm for 10 min at 4 °C. The resultant supernatants were stored at −80 °C for use in digestive enzyme assays. Spectrophotometric assays were used to determine enzymatic activities. The methods used to test the enzymatic activities were listed in Table 1. All enzymatic activities were expressed in units per milligram of protein (U mg$^{-1}$ prot).

## Statistical analysis

All statistical analyses were conducted using SPSS 20 software (IBM Corp., Armonk, NY, USA) with statistical significance set at α = 0.05. The data are expressed as mean ± standard deviation (SD). The results were initially tested for normality and homogeneity of variance using Shapiro-Wilk and Levene's tests, respectively. Two-way repeated-measures analysis of variance (ANOVA) was used to analyze filtration rate differences between groups (exposed and control) for each treatment over time. Enzymatic activities were compared using two-way ANOVA with group and treatment as factors. Tukey's honest significant difference (Tukey's Honestly Significant Difference (HSD)) test was used to determine differences within groups. One-way ANOVA was performed to determine

**Table 1 Spectrophotometric methods were used to assay enzymatic activities.**

| Enzyme | Substrate | Wavelength (nm) | Reference |
|---|---|---|---|
| Trypsin | N-benzoyl-L-arginine ethyl ester | 253 | *Inagami & Sturtevant (1960)* |
| Cellulase | Carboxymethylcellulose | 550 | *Fernández-Reiriz et al. (2001)* |
| Amylase | Starch | 660 | *Vega-Villasante, Nolasco & Civera (1993)* |
| Lipase | 2,3-dimercapto-1-propanol tributyrate | 570 | *Li et al. (2020)* |
| SOD | \ | 550 | *Paoletti et al. (1986)* |
| CAT | \ | 405 | *Regoli & Principato (1995)* |

differences between treatments for each group. Origin 2018 graphing software was used to create the diagrams (OriginLab Corporation, Northampton, USA).

# RESULTS

## Filtration rate

The exposed groups had lower chlorophyll *a* concentrations than the control groups starting from the first hour of the exposure experiment (Fig. 1). But at the end of the exposure experiment, the chlorophyll *a* concentrations were lower than 30 μg in all groups and treatments. The FR of *S. constricta* increased when exposed to high chlorophyll *a* concentrations of both toxic and non-toxic microalgae. The two-way ANOVA results indicated that group membership significantly ($P < 0.05$) affected treatments 3 and 4, while time significantly ($P < 0.05$) affected treatments 2, 3, and 4 ($P < 0.05$). There was no significant interaction between group and time for any treatment (Table 2). The FR of clams exposed to toxic microalgae was high during the first hour, decreased rapidly in the next 4 h, and remained at low levels during the rest of the experimental period (Fig. 2). The FR in the control group showed a similar trend, but was lower than in the exposed group during the first hour. The FR of clams exposed to toxic microalgae did not decrease as rapidly as that of the control group over the exposed period, but was higher than that of the control group at 6–12 h.

## Enzymatic activities

Trypsin and lipase activities were significantly ($P < 0.05$) affected by the concentration of chlorophyll *a* in the water (Fig. 3A). In the control group, trypsin activity under treatments 1 and 2 was significantly higher than that under treatments 3 and 4. Meanwhile in the exposed groups, trypsin activity under treatments 1–3 was significantly higher than that under treatment 4. Lipase activity under treatment 1 was significantly lower than that under treatments 2–4 in the control group, displaying an increasing trend with increasing chlorophyll *a* concentration. Meanwhile, lipase activity decreased with increasing chlorophyll *a* concentration in the exposed group (Fig. 3D), with significantly higher lipase activity under treatments 1–3 than under treatment 4. The lipase activity of the control group was significantly higher than that of the exposed group under treatments 3 and 4, whereas lipase activity under treatments 1 and 2 did not differ significantly between

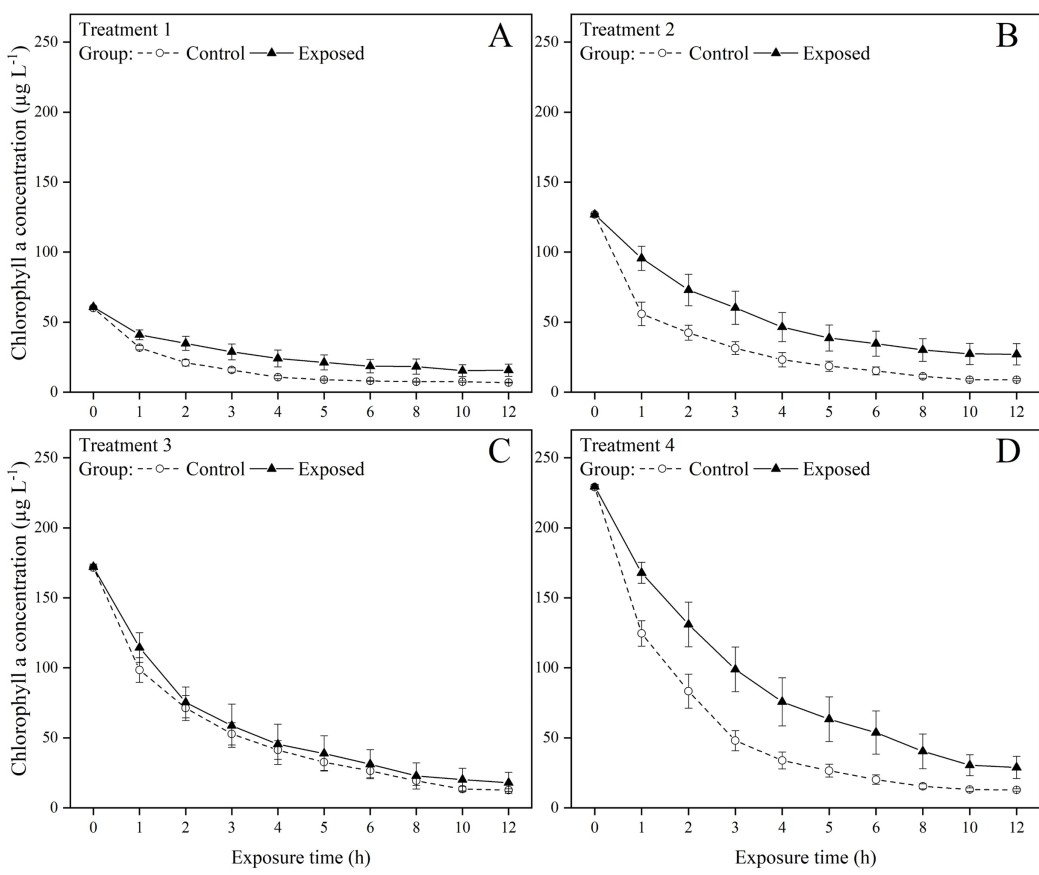

**Figure 1 Chlorophyll *a* concentration under (A) treatments 1 (chlorophyll *a* concentration: 60.2 ± 0.8 µg L⁻¹), (B) 2 (chlorophyll *a* concentration: 126.8 ± 1.4 µg L⁻¹), (C) 3 (chlorophyll *a* concentration: 171.9 ± 1.9 µg L⁻¹), and (D) 4 (chlorophyll *a* concentration: 229.2 ± 2.5 µg L⁻¹) during the 12 h exposure experiment.** Values are represented as means ± SE ($n = 6$).

groups. Cellulase and amylase activities did not differ significantly between groups and/or treatments (Figs. 3B and 3C).

SOD activity in the control group was significantly lower under treatments 3 and 4 than in the exposed group, whereas SOD activity under treatments 1 and 2 did not differ significantly between groups (Fig. 4A). Further, SOD activity did not differ significantly among treatments in the control group, whereas in the exposed group, it showed a non-significant increase with increasing chlorophyll *a* concentration. SOD activity in the exposed group under treatment 4 was significantly higher than that under treatment 1. CAT activity did not differ significantly between groups or treatments (Fig. 4B).

## DISCUSSION

### FR responds to toxic microalgal exposure

The feeding behavior of bivalves is affected by the consumption of toxic microalgae (*Zohdi & Abbaspour, 2019*). Indeed, bivalves have developed highly flexible feeding regimens in response to changes in the quantity and quality of suspended particles, enabling them to

**Table 2 Effects of group and sampling time point on filtration rate of S. constricta (two-way ANOVA results).**

| Factor | df | F | P |
|---|---|---|---|
| Treatment 1 | | | |
| Group | 1 | <0.001 | 0.987 |
| Time | 8 | 8.235 | 0.055 |
| Group & Time | 8 | 0.632 | 0.732 |
| Treatment 2 | | | |
| Group | 1 | <0.001 | 0.995 |
| Time | 8 | 24.946 | 0.012 |
| Group & Time | 8 | 0.341 | 0.900 |
| Treatment 3 | | | |
| Group | 1 | 6.901 | 0.025 |
| Time | 8 | 19.244 | 0.017 |
| Group & Time | 8 | 2.345 | 0.260 |
| Treatment 4 | | | |
| Group | 1 | 8.220 | 0.017 |
| Time | 8 | 176.409 | 0.001 |
| Group & Time | 8 | 1.135 | 0.509 |

optimize energy gains. *Nielsen et al. (2020)* suggested that blue mussel *Mytilus edulis* can reduced clearance rates of feeding when they exposed to DSP-toxic dinoflagellate. *Bauder et al. (2001)* investigated how bay scallops *Argopecten irradians* uptake, retain, and eliminate DSP toxins, reporting that bivalves can quickly accumulate DSP toxins and also detoxify them effectively. These studies suggest that some bivalve species may have greater tolerance to DSP toxins producing dinoflagellates in terms of their survival and feeding, but they still accumulate DSP toxins.

In the current study, the FR of *S. constricta* for toxic *P. cordatum* was lower than that for nontoxic *S. costatum* at all chlorophyll *a* concentrations at the beginning of the experiment, which suggested that toxic microalgae initially interfered with the feeding behavior of *S. constricta*. However, the FR for toxic microalgae did not decrease as rapidly as that for nontoxic microalgae at all chlorophyll *a* concentrations, likely because *S. constricta* gradually became used to feeding on toxic microalgae. Both the exposed and control groups were fed with the nontoxic diatom *S. costatum* prior to the exposure experiment. Therefore, clams in the exposed group may have required an adaptation period to the toxic dinoflagellate *P. cordatum*, which could explain the initially lower FR observed in exposed group. The FR of bivalves typically decreases with declining microalgal concentrations (*Sauvey et al., 2021*), which explains the rapid decrease in FR observed in the control group. Additionally, digestive enzymatic activities did not differ significantly between the control and exposed groups at low chlorophyll *a* concentrations (treatments 1 and 2), suggesting that this adaptation to toxic microalgae was not regulated by enzymatic activities at low toxic microalgae concentrations. Similar conclusions have been drawn in previous studies, indicating that the feeding activity of bivalves is not

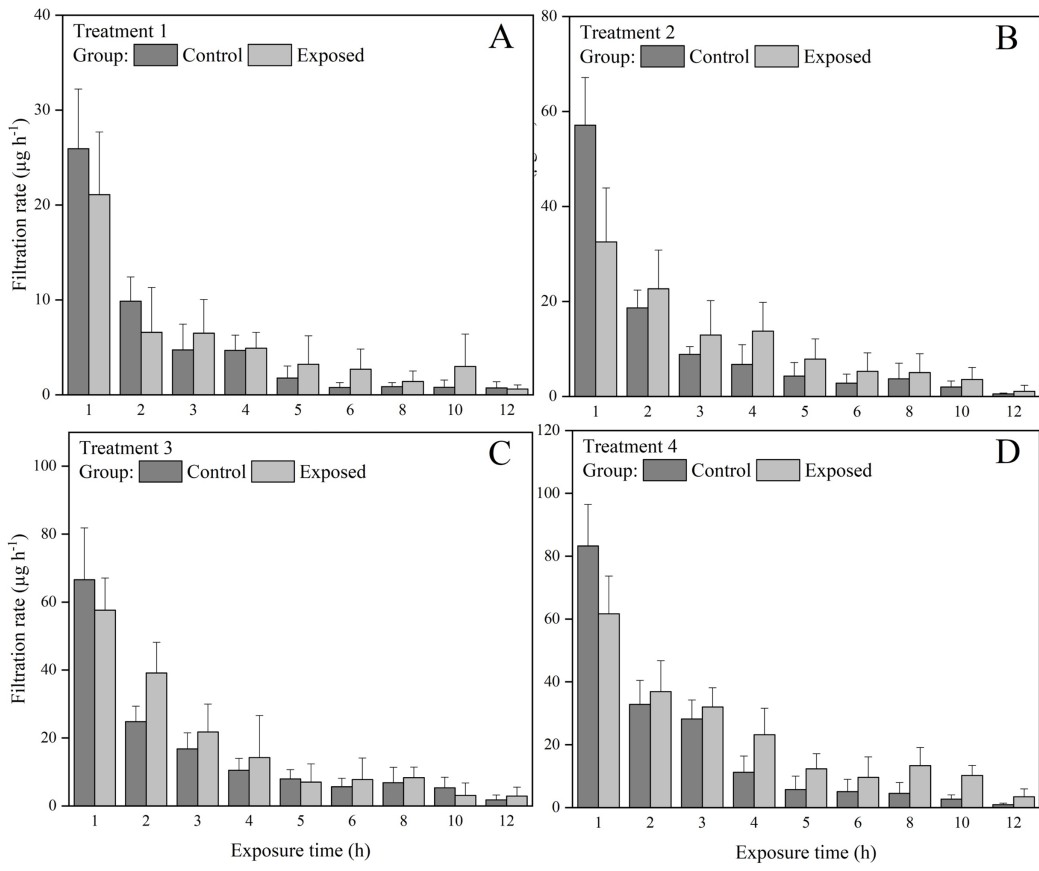

**Figure 2 Filtration rate (FR) of *S. constricta* under (A) treatments 1 (chlorophyll *a* concentration: 60.2 ± 0.8 µg L$^{-1}$), (B) 2 (chlorophyll *a* concentration: 126.8 ± 1.4 µg L$^{-1}$), (C) 3 (chlorophyll *a* concentration: 171 ± 1.9 µg L$^{-1}$), and (D) 4 (chlorophyll *a* concentration: 229.2 ± 2.5 µg L$^{-1}$) during the 12 h exposure experiment.** Values are represented as means ± SD ($n$ = 6).

affected by toxic microalgae at certain concentrations (*Bauder et al., 2001*; *McGuire et al., 2025*). Since the chlorophyll *a* concentrations in treatments 1 and 2 were comparable to those observed during natural HAB events, *S. constricta* may maintain normal feeding function in natural environments during such blooms.

Only high chlorophyll *a* concentrations (treatments 3 and 4) caused significant differences in FR between the control and exposed groups. This result illustrates that *S. constricta* can tolerate low concentrations of toxic microalgae. Moreover, the chlorophyll *a* concentration in the natural environment during HABs is usually less than 20 µg L$^{-1}$ (*Wei, Tang & Wang, 2008*; *Stauffer et al., 2019*), which is much lower than the chlorophyll *a* concentrations used in the current experiment. Therefore, *S. constricta* could be considered a potential bioremediation species for HABs in natural seawater where the chlorophyll *a* concentration is lower. *Zhang et al. (2022)* previously reported that a high razor clam stocking density could reduce the nontoxic phytoplankton biomass and net primary production in mariculture ponds with swimming crabs and shrimp. Similarly, *Jiang et al. (2019)* found that cultivated oysters control phytoplankton blooms in natural

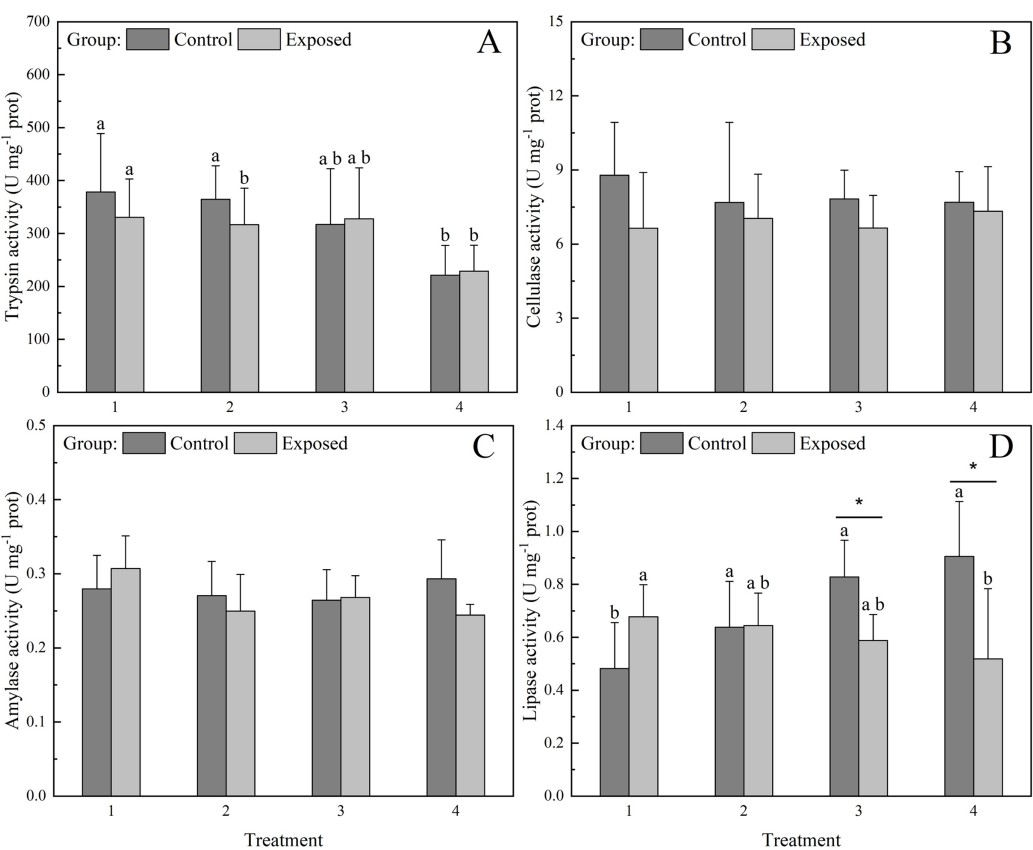

**Figure 3 Activities of digestive enzymes (A) Trypsin, (B) Cellulase, (C) Amylase, and (D) Lipase in the digestive glands of *S. constricta* after the 12 h exposure experiment.** Values are represented as means ± SD ($n = 6$). Different letters indicate significant differences among treatments within each group (Tukey's HSD test, $P < 0.05$). Asterisks indicate significant differences between groups within each treatment (one-way ANOVA, $P < 0.05$).

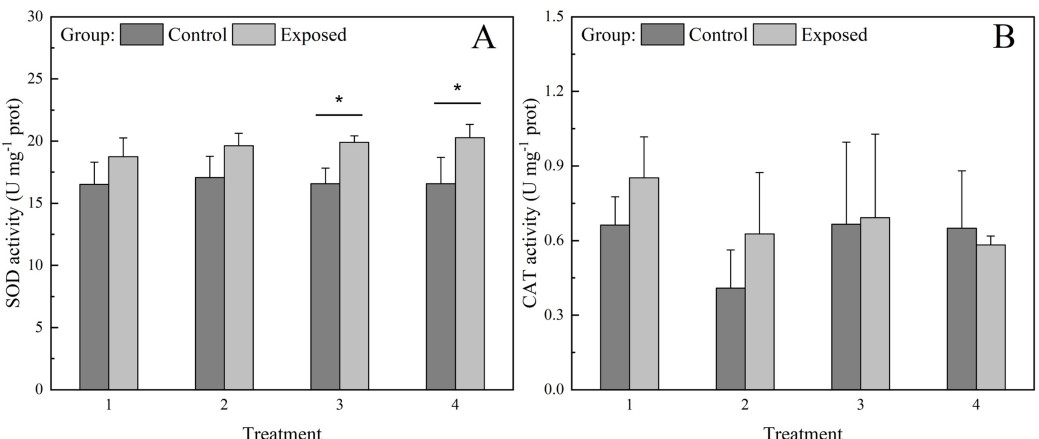

**Figure 4 Activities of antioxidant enzymes (A) superoxide dismutase (SOD) and (B) catalase (CAT) (B) in the digestive glands of *S. constricta* after the 12 h exposure experiment.** Values are represented as means ± SD ($n = 6$). Different letters indicate significant differences among treatments within each group (Tukey's HSD test, $P < 0.05$). Asterisks indicate significant differences between groups within each treatment (one-way ANOVA, $P < 0.05$).

waters of Xiangshan Bay, China. Since *S. constricta* is one of the major polyculture species used in East Asian aquaculture, along with shrimp and crab (*Xie, Jiang & Yang, 2011*; *Guan et al., 2020*; *Zhang et al., 2022*), our study demonstrates that *S. constricta* could be used to efficiently reduce nontoxic as well as toxic phytoplankton density.

## Digestive enzymatic activities respond to exposure to toxic microalgae

Digestive enzymatic activities, as important indicators of nutritional status, reflect the digestion performance of bivalves to a certain degree (*Albentosa & Moyano, 2008*). Previous studies have reported that microalgal abundance and species influence FR and digestive enzymatic activities of bivalves (*Galimany et al., 2020*; *Sauvey et al., 2021*). In the current study, higher toxic phytoplankton concentrations (treatments 3 and 4) caused statistically significant changes in lipase activity. Reverse trends in lipase activity between the control and exposed groups confirmed that lipase inhibition in the exposed group was correlated with the toxicity of *P. cordatum*. Lipase activity of *S. constricta* is significantly associated with environmental factors such as light intensity and pH (*Liu et al., 2021*; *Liang et al., 2022*). Therefore, the present results support that lipase is more sensitive to toxic microalgae than the other digestive enzymes we analyzed in this study.

Eukaryotic phytoplankton, such as diatoms and dinoflagellates, predominantly accumulate neutral lipids, mainly in the form of triacylglycerols (TAGs) (*Becker et al., 2018*). TAGs are more effective energy stores than carbohydrates because they contain more chemical energy per mole of carbon and larger quantities can be stored inside the cell (*Berg et al., 2015*). Thus, lipids in microalgae are an important energy source for bivalves. The digestive capability and dietary preference of bivalves are typically closely related (*Li et al., 2020*). Therefore, the observed increase in lipase activity from low to high chlorophyll *a* concentrations in the control group may suggest a digestive preference of *S. constricta* for lipids. Lipase activities of clams in the exposed group were possibly inhibited by toxic microalgae, which may have resulted in insufficient energy absorption by *S. constricta*.

Trypsin activity decreased significantly from low to high chlorophyll *a* concentrations in both the exposed and control groups, suggesting that trypsin inhibition was more likely caused by high density rather than microalgal toxicity. The microalgal densities under all treatments were higher than those during HABs, which usually occur in natural seawater (*Wei, Tang & Wang, 2008*; *Stauffer et al., 2019*). Indeed, the highest trypsin activity of *S. constricta* may be achieved at lower phytoplankton concentrations than were used in the present study. Several studies have demonstrated the importance of proteases, including trypsin, as key enzymes for feed utilization and growth due to their role in protein digestion processes (*Rungruangsak-Torrissen et al., 2006*; *Albentosa & Moyano, 2008*; *Klomklao, 2008*). However, distinct proteases respond differently to environmental factors (*Korez, Gutow & Saborowski, 2019*). Other protease activities may increase along with increasing microalgal density while trypsin activity decreases. The activities of digestive enzymes, except lipase, did not differ significantly between the exposed and control groups. These results support that the digestive process of *S. constricta* functions normally when exposed to high-density toxic phytoplankton.

## Antioxidant enzymes activities respond to exposure to toxic microalgae

SOD and CAT are key antioxidant enzymes that determine the effectiveness of an antioxidant system. Studies have shown that bivalves produce antioxidant enzymes to protect against oxidative stress induced by absorbed microalgal toxins (*Shumway, 1990*; *Vidal et al., 2014*; *Ye et al., 2022*). The results of the present study revealed that SOD activity was higher at higher chlorophyll *a* concentrations (treatments 3 and 4). Higher SOD activity has also been observed in *S. constricta* following short-term exposure to elevated levels of suspended solids (*Yang et al., 2017*). Higher SOD activity at highly toxic microalgae concentrations may be a consequence of the enzyme's detoxifying role.

SOD and CAT activities in the digestive gland of *S. constricta* were much lower than those previously reported in *M. trossulus*, *C. gigas* and *Mactra chinensis* (*Istomina et al., 2021*). *S. constricta* is a typical burrowing bivalve species that usually buries itself in the sediments of mudflats. Although the activities of antioxidant enzymes in other razor clam species have rarely been reported, a previous study demonstrated that burrowing bivalves generally have low metabolic rates and antioxidant system activity, reduced mitochondrial function, and less accumulation of compounds that cause cellular damage (*Philipp, Strahl & Sukhotin, 2012*). Furthermore, bivalves buried in sediment can retract their siphons and remain in a state of anoxia for several days (*Istomina et al., 2021*). Nevertheless, SOD and CAT activities were low even with no sediment in which to bury throughout the exposure experiment, suggesting that *S. constricta* may have limited antioxidant ability compared to non-burrowing bivalves. However, sheltering in mudflat sediments may protect *S. constricta* against toxic microalgae during HABs in the natural environment.

## Bioremediation function of *S. constricta*

The FR of *S. constricta* increased with higher chlorophyll *a* concentration. Similarly, some bivalve species removed more toxic microalgae when they were exposed to higher cell concentration (*Galimany, Lunt & Freeman, 2021*). However, high concentration of toxic microalgae impacted the physiological process of *S. constricta* by inhibiting lipase activity and inducing SOD activity. The intensive HABs have caused tremendous economic loss in the aquaculture industry globally (*Trottet et al., 2021*). While the long-term effects of toxic microalgae on *S. constricta* remain unclear, the short-term exposure in this study suggests that *S. constricta* may not tolerate extremely high concentrations of toxic microalgae. However, it appears capable of withstanding most HABs typically found in natural seawater. As an important commercial species in East Asia, our results suggested the harmful algae polluted sea area is not a proper place for *S. constricta* aquaculture.

Bivalves filter water and particles, and create suitable habitat for other species (*van der Schatte Olivier et al., 2020*). *S. constricta* is considered an aquaculture bivalve, but its role as a water purifier has not been adequately explored, although a previous study has shown that *S. constricta* can promote nutrient recycling in eutrophic waters (*Zhao et al., 2019*). Many studies have reported non-selective feeding behavior in bivalves, as indicated by the similar seasonal patterns of microalgae composition observed in both seawater and bivalve stomach contents (*Kamermans, 1994*; *Rouillon et al., 2005*; *Houki, Ozaki & Sano, 2025*).

Outdoor large-scale cultivation of *S. constricta* may reduce the abundance of both toxic and non-toxic microalgae, potentially leading to a decline in local primary productivity (*Smaal et al., 2013*) and alterations in the food web (*Vaughn & Hoellein, 2018*). Therefore, further research is needed to confirm the bioremediation function of *S. constricta*.

## CONCLUSIONS

Short-term exposure to the toxic dinoflagellate *Prorocentrum cordatum* significantly affected the filtration rate (FR) and physiological processes of the Chinese razor clam *Sinonovacula constricta*. At high concentrations of toxic microalgae, digestive lipase activity was notably inhibited. While superoxide dismutase (SOD) activity was upregulated in response to higher concentrations of toxic microalgae, the overall antioxidant enzymatic activity in *S. constricta* was lower compared to other bivalves reported in the literature. These findings suggest that *S. constricta* can adapt to toxic microalgae through changes in feeding behavior, as well as modifications in digestive and antioxidant enzymatic activities. This adaptive response supports the potential of *S. constricta* as a bioremediation species for mitigating harmful algal blooms (HABs). However, the full extent of the physiological mechanisms underlying these responses remains unclear. Future studies incorporating metabolomic and transcriptomic analyses could offer deeper insights into how bivalves, such as *S. constricta*, cope with harmful microalgae.

## ACKNOWLEDGEMENTS

We are grateful to Ningbo Academy of Oceanology and Fishery for providing the test site.

### Funding
This study was supported by the National Key Research and Development Program of China (Grant No. 2021YFC3101702). The funders had no role in study design, data collection and analysis, decision to publish, or preparation of the manuscript.

### Grant Disclosures
The following grant information was disclosed by the authors:
National Key Research and Development Program of China: 2021YFC3101702.

### Competing Interests
The authors declare that they have no competing interests.

### Author Contributions
- Yanbin Tang performed the experiments, analyzed the data, prepared figures and/or tables, and approved the final draft.
- Zhibing Jiang performed the experiments, prepared figures and/or tables, and approved the final draft.
- Yibo Liao performed the experiments, authored or reviewed drafts of the article, and approved the final draft.

- Lu Shou conceived and designed the experiments, authored or reviewed drafts of the article, and approved the final draft.
- Jiangning Zeng conceived and designed the experiments, authored or reviewed drafts of the article, and approved the final draft.
- Rongliang Zhang performed the experiments, prepared figures and/or tables, and approved the final draft.
- Chenghua Li conceived and designed the experiments, authored or reviewed drafts of the article, and approved the final draft.

## Data Availability

The raw data are available in the Supplemental File.

## Supplemental Information

Supplemental information for this article can be found online at http://dx.doi.org/10.7717/peerj.20072#supplemental-information.

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
