# Peer review of "Toxic dinoflagellate Prorocentrum cordatum affects the filtration rate and enzymatic activities of Chinese razor clam (Sinonovacula constricta)"

_PeerJ, doi:10.7717/peerj.20072_

## Round 0.1 · original submission · Major Revisions

· Academic Editor

Major Revisions

Dear authors,

I have now received three reviews of your manuscript, two of which recommend acceptance after revision and one of which argues against publication in this form. In view of these contradictory recommendations, I urge you to carefully consider all the reviewers' suggestions, as there are obviously some issues that need to be addressed before acceptance. What all three reviewers have in common is that the conclusions are not fully supported by the findings and that the discussion needs to be toned down with this in mind. However, the topic is considered interesting, with the potential for a significant contribution to the field.
Based on this feedback and my personal opinion, I invite you to revise your manuscript in line with the reviews received.

In particular, please consider the requested changes to the abstract and introduction, clarification of methodological issues (raised by reviewers 2 and 3), and reformulation of the conclusions to better reflect the validity of the results.
Please include in the revised version of your manuscript a detailed response to the comments received (point by point) with clear answers and references to corrections.

Kind regards,
Olja Vidjak

Reviewer 1 ·

Basic reporting

This was a very good paper describing the effects of exposure to toxic dinoflagellates on the razor clam feeding rate and enzyme activity. The writing was mostly clear; I have made minor grammatical and English suggestions throughout. The figures and tables were appropriate, but the font/characters in some figure and table legends did not translate and should be corrected. In general, conclusions were appropriate, but I have noted instances where the connection between results and interpretation was weak. In general, this paper is a very important step in furthering our understanding of razor clam responses to toxic algae exposure and lays a foundation for future research on the possibility of using razor clams as a bioremediation tool. Keep up the excellent work!

Please see detailed comments in the attached PDF.

Experimental design

-

Validity of the findings

In general, conclusions were appropriate, but I have noted instances where the connection between results and interpretation was weak. Please see detailed comments in the attached PDF.

Additional comments

Please see detailed comments in the attached PDF.

Annotated reviews are not available for download in order to protect the identity of reviewers who chose to remain anonymous.

Reviewer 2 ·

Basic reporting

The authors do not demonstrate a mastery of the material necessary to engage in this research. While many exciting scientific insights have come from all levels of expertise, this current project feels like an introductory science project that suffers from serious flaws.

The authors repeatedly use the word 'adapt' when acclimate is accurate. This may sound trivial, but the difference is much greater than just semantics. There was also very little background on why the current bivalve-HAB was chosen. Do they overlap in nature? Why choose them? Do the clams even have receptors to make them susceptible to these specific toxins? There is no justification for this study. They say that HABs are also known as 'Red Tides'. Also inaccurate. Many of the references are used over and over again, and are not the best of most recent.

Experimental design

The design has many flaws. There is one treatment and one control group. Yes, there are 24 in each, but there is no replication. It says dead clams were removed and replaced? Did they die from the food? Also, we have no idea if the overall sizes (cumulative) of the clams from each group are, or even the same. It wasn't clear if they scaled filtration rates. Algae measurements were taken using Chl a, also the least preferred method. It measures chlorophyll disappearance, not filtration rate. Cell counts would be much more useful.

Validity of the findings

Unfortunately, the filtration rate findings are hard to interpret due to the flaws of the design.

Additional comments

Even if the data were sound, the conclusion that these clams could be used for bio remediation of toxic blooms is weak. A short-term assay cannot be extrapolated to a population-level effect for the algae. There was no data or models that showed the clams could sustain a filtration rate high enough to affect the algae's growth. Plus, long-term exposure to the clam is unknown.

Reviewer 3 ·

Basic reporting

The article is well-written in general. It brings important information for the scientific community and should be accepted for publication. I suggested major revisions because the conclusion is not supported by the results. I wrote some suggestions that, if accepted, I believe will help the authors reformulate their conclusion.
Due to the increasing frequency and length of harmful algal blooms and the knowledge gap on their effects on filter feeders’ overall health, this manuscript brings relevant information. The abstract is well-written and clear. However, the authors should mention how the negative effect on the antioxidant defenses when exposed to elevated levels of toxic microalgae could affect bivalve populations (this is a nice conclusion too because many works assume that toxic dinoflagellates do not have a negative impact on shellfish but as this study showed in the short-term and using adults, toxic dinoflagellates might affect the resilience of important commercial species, such as the Chinese razor clam). I am also missing in the abstract the effect on the enzymatic activities when exposed to the non-toxic algae. In addition, the last sentence is not supported by the results. How can these bivalves be used as a bioremediation species if, in the short term, increasing concentration of toxic dinoflagellates inhibits lipase activity and affects their antioxidant defenses for detoxification? The study was carried out for 12 hours with adult individuals (1 year old). However, long-term exposure to toxic algae could end up having an even more deleterious effect on the razor clams (especially in early stages such as larvae and juveniles).
The introduction showed the overall context, but more information on the neutral/positive effect of toxic microalgae on shellfish should also be included. The results and graphs should be improved, see comments below. The raw data was supplied by the authors.
Line 33. Why do you think that the filtration rates decreased that much with non-toxic microalgae? Are the filtration levels normal when compared with previous studies?
Line 37. Include toxic before microalgal. What about the enzymatic responses of bivalves fed with nontoxic microalgae? Include a brief sentence.
Line 62. The authors should also include some studies that found no negative effect of toxic microalgae on bivalves, because, as stated, bivalves also consume toxic microalgae.

Experimental design

The research question is defined and relevant, and the methods are described in detail. Instead of “experimental group,” I suggest using “exposed group” (it is more correct because everything was tested experimentally).
Line 107. Substitute “calms” for “clams.”
Lines 110-111. How did you establish the different concentrations tested? Are they based on the concentrations found in the field under natural conditions during HABs? Give further information, please.
Line 112. Did you use only one clam per replicate? Did you have any mortality during the experiment? In that case, what did you do? Because there was only one clam per replicate, if there was any mortality during the experiments, this information should be included.
Lines 111-114. The sentence “The cultivated microalgae were diluted…S. costatum” should be replaced before the sentence at line 108 that starts with “The experimental and control groups… (treatment 4)”.
Line 114. Include after P. cordatum “(exposed group)” and after S. costatum “(control group).

Validity of the findings

The conclusions are stated, and the discussion is relevant. However, the statistical results are not robust enough to support the conclusion that S. constricta is a bioremediation species for mitigating the impacts of harmful algal blooms. In addition, the concentration of toxic microalgae during harmful algal blooms to justify the levels used during this study should appear before the discussion, and the bioaccumulation of this toxic microalgae on the sediment should also be considered.
Lines163-174. Change “experimental group” to “exposed group” throughout the manuscript. The results section could be improved to show if there were differences among treatments because it is difficult to follow. You could refer to “exposed to toxic microalgae” and “control” to facilitate its reading.
Line 191. Change “it increased slowly”… was it significant?
In Figure 1, I would like to see the complete standard deviation graph and not only half because it seems that there are no differences among groups in any of the treatments by the end of the experiments (you could also use standard error to avoid overlapping). In the figure caption, you could also include the concentrations of microalgae used for each treatment in brackets to help with the visualization of the data.
In Figure 2, it will be facilitated if you include the significance among the groups for the different exposure times with letters or asterisks (as in the rest of the figures).
In Figures 3 and 4, if the results are not significantly different among treatments, you should include no letters as it is right now; it is very confusing.
Line 215-216. Are there any other articles that have found similar results under low concentrations of toxic microalgae? What about the concentrations found in the field under natural conditions during HABs? The concentration found in the field under HABs is close to which of the levels tested?
Another factor that the authors should consider is that at the beginning all individuals were fed with the same microalgae sp., and then after starving them for some days you changed the diet of the “experimental/exposed group” and this could also explain the lower FR for the clams exposed to toxic microalgae during the first two hours (they were adapting to a different type of food). This possibility has to be discussed.
Line 250. How could the lipase inhibition in the exposed group affect the clams’ performance and resilience to environmental changes? Please include further information.
Line 283. Which bivalve species? Have there been any studies on other species of razor clams?
Lines 300-303. Razor clams can decrease toxic microalgae, but with negative consequences for the organisms. Therefore, in the long term and under elevated levels of toxic microalgae, this species might not be resilient enough to bioremediate HABs. Also, due to its high filtering capacity, extensive aquaculture of this species “for bioremediation” could end up impacting other species because they filter everything, including the nontoxic microalgae present, reducing primary production and the food availability for other species. Please, consider including this point in the discussion.
Lines 312-324. The species names are not in italics; please change and verify the rest of the manuscript.
Supplementary materials.
In the “Chlorophyll a” and “enzymatic activities” files, correct the spelling of “treatemnt” to “treatment”. In the enzymatic activities file, also correct the spelling of “contril” to “control”. In all files, change “experimental” to “exposed”.

---

## Round 0.2 · Minor Revisions

· Academic Editor

Minor Revisions

Dear authors,

Thank you for completing the first round of reviews and for considering the reviewers' suggestions. The reviewers declined the invitation to re-review, but at the suggestion of one of them, I consulted another reviewer with special expertise in the field of toxic microalgae, who suggested some minor changes to improve the quality and visibility of your manuscript. As I consider the points raised to be valid, I invite you to make an additional effort to take this review into account. Please include in the revised version of your manuscript a detailed response to the comments received (point by point) with clear answers and references to corrections.

Please also ensure that;
- maintain consistency of citations in the text (e.g. lines 304 and 314)
- improve clarity in line 327, the sentence should read as follows: S. constricta is considered an aquaculture bivalve, but its role as a water purifier has not been adequately explored, although a previous study has shown that S. constricta can promote nutrient recycling in eutrophic waters (Zhao et al., 2019).
- L 327 begins with ...Many studies… but only 1 is cited, please add more citations for this statement.
- L337 and 327 - please provide the full name of both species (first mention in the section)

Kind regards,
Olja Vidjak

Reviewer 4 ·

Basic reporting

Introduction:
Comment: oversimplified and lacks citing more topic-specific literature.

Line 76- 78 “Mytilus edulis and Crassostrea virginica can actively reject or avoid toxic cells (Rosa et al., 2017), while M. galloprovincialis ingests and tolerates toxins, incorporating them into its tissues (Wu et al., 2022). “

Comment: Rosa et al. studied the physical and chemical surface properties of various microalgae used as food for suspension-feeding bivalves, while Wu et al. investigated the effects of paralytic shellfish toxin (PST) profiles and concentrations on variations in the condition index of bivalves under natural bloom conditions. The results of these two studies are not directly comparable. However, both tested multiple bivalve species. Please rephrase your sentence. Moreover, since Wu et al. focused on a PSP-producing microalgae, and your study involves DSP-producing species, it would be more appropriate to identify and cite literature specifically addressing the effects of DSP-producing microalgae.

Line 94-96 “The results of this study will improve our understanding of the feasibility of using S. constricta to mitigate HAB events.”

Comment: How will the results of this study mitigate HAB events? Please explain.

Discussion
Comment: The type of toxin produced by the microalgae is highly relevant in determining its effects on bivalves, including impacts on filtration rates. However, the authors overlook this critical distinction and frequently reference literature on PSP-producing microalgae, rather than focusing on studies involving DSP-producing species. This weakens the discussion, as DSP and PSP toxins differ significantly in their chemical nature and physiological effects. DSP toxins are lipophilic and tend to accumulate and persist longer in bivalve tissues. In contrast, PSP toxins are hydrophilic. The discussion should be revised to reflect these differences and supported with appropriate literature specific to DSP-producing microalgae.
Moreover, the analysis of how much okadaic acid accumulated in the bivalves in each treatment will help in the interpretation of the enzymatic activities results.

Experimental design

no comment

Validity of the findings

no comment

---

## Round 0.3 · Minor Revisions

· Academic Editor

Minor Revisions

Dear author, thank you for considering the reviewer's suggestions. As the reviewer considered this to be a minor revision, I will not be requesting a re-review, but have carried out a final check of all changes before acceptance. You will find my comments on the manuscript attached to this message, and I ask you to consider them to finalise the revision process.

In particular, I would like to ask you to replace some of the newly included references with more appropriate ones, as they do not fully reflect the points made (Shen et al. and Goncharenko et al.). The remaining comments are minor suggestions and linguistic corrections.

Thank you for your co-operation,
Olja

---

## Round 0.4 · accepted · Accept

· Academic Editor

Accept

Dear authors, thank you for finalising the review, and consider this a final version ready for acceptance and publication.

Congratulations and thank you for choosing PeerJ as your publication platform.

Kind regards,
Olja